# Impact of Body Mass Index on the Survival of Patients with Sepsis with Different Modified NUTRIC Scores

**DOI:** 10.3390/nu13061873

**Published:** 2021-05-30

**Authors:** Yi-Hsuan Tsai, Chiung-Yu Lin, Yu-Mu Chen, Yu-Ping Chang, Kai-Yin Hung, Ya-Chun Chang, Hung-Cheng Chen, Kuo-Tung Huang, Yung-Che Chen, Yi-Hsi Wang, Chin-Chou Wang, Meng-Chih Lin, Wen-Feng Fang

**Affiliations:** 1Division of Pulmonary and Critical Care Medicine, Department of Internal Medicine, Kaohsiung Chang Gung Memorial Hospital, Chang Gung University College of Medicine, Kaohsiung 83301, Taiwan; yhtsai@cgmh.org.tw (Y.-H.T.); chiungyu@cgmh.org.tw (C.-Y.L.); blackie@cgmh.org.tw (Y.-M.C.); b9002087@cgmh.org.tw (Y.-P.C.); redrosahung@yahoo.com.tw (K.-Y.H.); y7817@cgmh.org.tw (Y.-C.C.); chc1106@cgmh.org.tw (H.-C.C.); jelly@cgmh.org.tw (K.-T.H.); yungchechen@yahoo.com.tw (Y.-C.C.); yihsi@cgmh.org.tw (Y.-H.W.); ccwang5202@yahoo.com.tw (C.-C.W.); mengchih@cgmh.org.tw (M.-C.L.); 2Graduate Institute of Clinical Medical Sciences, Chang Gung University, Taoyuan 33302, Taiwan; 3Department of Nutritional Therapy, Kaohsiung Chang Gung Memorial Hospital, Kaohsiung 83301, Taiwan; 4Department of Respiratory Therapy, Kaohsiung Chang Gung Memorial Hospital, Chang Gung University College of Medicine, Kaohsiung 83301, Taiwan; 5Department of Respiratory Care, Chang Gung University of Science and Technology, Chiayi 61363, Taiwan

**Keywords:** sepsis, nutrition, survival, underweight

## Abstract

Nutritional status affects the survival of patients with sepsis. This retrospective study analyzed the impact of body mass index (BMI) and modified nutrition risk in critically ill (mNUTRIC) scores on survival of these patients. Data of 1291 patients with sepsis admitted to the intensive care unit (ICU) were extracted. The outcomes were mortality, duration of stay, ICU stay, and survival curve for 90-day mortality. Logistic regression analysis was performed to examine the risk factors for mortality. Cytokine and biomarker levels were analyzed in 165 patients. The 90-day survival of underweight patients with low mNUTRIC scores was significantly better than that of normal-weight patients with low mNUTRIC scores (70.8% vs. 58.3%, respectively; *p* = 0.048). Regression model analysis revealed that underweight patients with low mNUTRIC scores had a lower risk of mortality (odds ratio = 0.557; *p* = 0.082). Moreover, normal-weight patients with low mNUTRIC scores had the lowest human leukocyte antigen DR (HLA-DR) level on days 1 (underweight vs. normal weight vs. overweight: 94.3 vs. 82.1 vs. 94.3, respectively; *p* = 0.007) and 3 (91.8 vs. 91.0 vs. 93.2, respectively; *p* = 0.047). Thus, being underweight may not always be harmful if patients have optimal clinical nutritional status. Additionally, HLA-DR levels were the lowest in patients with low survival.

## 1. Introduction

Sepsis is characterized by life-threatening organ dysfunction caused by dysregulated host response to infection [1]. According to an Asian transnational study, the in-hospital mortality rate of patients with sepsis was reported to be 44.5% [2]. Previous studies have identified risk factors for mortality, such as comorbidities [3,4], poor nutrition [5,6], abnormal clinical parameters [7,8], and impaired immunity [3,9,10], in patients with sepsis. Identification of simple phenotypes in patients with strong survivability is difficult because sepsis is highly heterogeneous [11]. Nevertheless, identification of phenotypes or sub-phenotypes of patients with sepsis with different survival rates would be promising for clinical risk stratification.

Nutritional status is likely to be a phenotype of sepsis [12,13]. Previous studies have reported that half of hospital in-patients have nutritional problems [14,15,16]. Malnutrition in patients with sepsis may increase the risk of mortality [17,18,19,20,21,22,23,24] and duration of ventilator dependence [25]. In these studies, body mass index (BMI), a simple variable calculated using the following formula: BMI = kg/m^2^ (kg: a person’s weight in kilograms; m^2^: person’s height in meters squared), was commonly used to indicate the nutritional status of the general population. However, the association between BMI and patient outcomes remains controversial. Overweight and obesity may be associated with lower mortality than normal weight [23,26,27,28,29,30,31,32], owing to the higher metabolic reserves in acute catabolic illnesses among individuals with higher BMIs. In addition, adipose tissue secretes anti-inflammatory mediators, such as leptin and tumor necrosis factor (TNF) receptor 2 [33]. In contrast, some studies reported that overweight and obesity increased the risk of mortality [20,21,22,34,35], especially in patients with cardiovascular disease [20,35], diabetes, and kidney disease [20]. Other studies have shown no significant differences in the risk of mortality between obese and normal-weight critically ill patients [36,37,38,39]. These controversial results imply that true nutritional status cannot be represented by simple BMI-related stratifications.

Therefore, scores have been developed to evaluate the nutritional status of critically ill patients [6,12,13]. The modified nutrition risk in critically ill (mNUTRIC) score simplifies the NUTRIC score by excluding the interleukin 6 (IL-6) variable. It includes age, the acute physiology and chronic health evaluation II (APACHE II) score, the sequential organ failure assessment (SOFA) score, number of comorbidities, and duration of admission to the intensive care unit (ICU). This score has been verified for the nutritional evaluation of critically ill patients. A study showed that patients with higher mNUTRIC scores had significantly higher mortality rates [6]. However, the mNUTRIC score does not include the BMI domain and is more complex than the BMI.

In addition, sepsis-induced cytokine changes related immune dysfunction have an impact on clinical outcomes [9]. Fajgenbaum and June’s review revealed that IL-6, IL-1, IL-17, TNF, and interferon gamma (IFN-γ) were released during a cytokine storm. IL-10 and IL-1 receptor antagonist (IL-1RA) are negative regulators of cytokine storms [40]. In our previous studies, lower human leukocyte antigen DR (HLA-DR) expression was associated with higher mortality due to nutritional insufficiency [5]. Patients had a higher risk of mortality when IL-6 and TNF-α levels were higher [9]. However, it remains unclear how cytokine levels differ among patients with sepsis who have different nutritional statuses. Because nutritional status affects the outcomes of patients with sepsis, it is important to identify the role of cytokines in nutrition and sepsis.

In this study, we proposed a nutritional phenotype based on complete nutritional status that consists of BMI and the mNUTRIC score. To examine how this nutritional phenotype affects the survival of patients with sepsis, this study analyzed the impact of BMI and the mNUTRIC score on the survival of these patients. Based on the conceptual framework mentioned above, this study aimed to determine the following: (a) survival of patients with sepsis classified into different BMI and mNUTRIC score groups, (b) impact of the mNUTRIC scores on the survival of patients with sepsis who had different BMIs, (c) impact of the BMI on the survival of patients with sepsis who had different mNUTRIC scores, and (d) the cytokine levels linked with different nutritional phenotypes. Specifically, we used multiple domains and steps to identify the heterogeneity of nutritional status and clarify its impact on the survival of patients with sepsis. Biological data analyses were performed to confirm the results. Our results may help in further understanding whether the nutritional phenotype influences survival of patients with sepsis, which could help improve their care and treatment.

## 2. Materials and Methods

### 2.1. Study Participants

This retrospective study enrolled adult patients with sepsis who were admitted to medical ICUs (34 beds) at Kaohsiung Chang Gung Memorial Hospital between August 2013 and January 2017. Patients with sepsis who were admitted to the medical ICU between January 2020 and August 2020 were included in the validation cohort. All participants who fulfilled the criteria of the Third International Consensus Definitions for Sepsis and Septic Shock (Sepsis-3) [1] were screened. Clinical parameters, including age, sex, BMI, APACHE II score, Charlson comorbidity index, SOFA score, NUTRIC score, comorbidities, infection site, laboratory data, and day of mortality, were obtained from medical records. In addition, the study enrolled 165 patients who agreed to undergo blood sampling during ICU hospitalization. All patients or family members provided written informed consent for the use of residual blood samples. The study design was approved by the Institutional Review Board (ID: 202001696B0C501) of Chang Gung Memorial Hospital.

### 2.2. Definitions

This study used the mNUTRIC score and BMI to evaluate the nutritional status of the participants. The NUTRIC score includes six parameters: age, APACHE II score, SOFA score, number of comorbidities, IL-6 level, and hospital stay before admission to the ICU [12]. The mNUTRIC score includes all these parameters except IL-6 [6] (Table 1). In our cohort, the median mNUTRIC score was 6, mode was 6, and mean was 6.01. Thus, we chose 6 as the cut-off value for classification into low and high mNUTRIC score groups. Participants with mNUTRIC scores of ≥6 were classified into the high mNUTRIC score group, and those with mNUTRIC scores of <6 were included in the low mNUTRIC score group. BMI was defined as the value obtained by dividing the weight of the participant in kilograms by the square of the height in meters. This was calculated when the patients were admitted to the ICU. BMI was categorized into three groups: underweight (<18.5 kg/m^2^), normal-weight (18.5–24.9 kg/m^2^), and overweight (≥25.0 kg/m^2^).

### 2.3. Biomarkers

The levels of HLA-DR monocyte expression and cytokines, including IL-6, IL-1RA, IL-10, IL-17, TNF-α, and IFN-γ in the plasma, have been described in our previous reports [3,5,7,9]. All data were collected after admission to the ICU on days 1, 3, and 7.

### 2.4. Outcome

The main outcome of this study was survival rate. We compared 7-day, 14-day, 28-day, and 90-day mortality rates between patients in the different BMI and mNUTRIC score groups. The 90-day mortality rate was used to construct the survival curves. The secondary outcomes were length of stay in the hospital and ICU.

### 2.5. Statistical Analyses

The data analysis was performed using SPSS software (version 22.0; IBM Corp., Armonk, NY, USA). The high and low mNUTRIC scores were compared in terms of demographic characteristics, baseline characteristics, and outcomes. The Mann-Whitney U test was used for continuous variables and chi-square test was used for categorical variables. The survival curve was constructed according to the 90-day mortality that was determined based on the mNUTRIC score. Subsequently, the demographic characteristics, baseline characteristics, and outcomes were compared across the underweight, normal-weight, and overweight groups. The Kruskal-Wallis test was used for continuous variables and chi-square test was used for categorical variables. The survival curve was constructed according to the 90-day mortality that was determined based on the different BMI groups. For the three BMI groups, we created a survival curve from the 90-day mortality based on the mNUTRIC scores.

In the low and high mNUTRIC score subgroups, we compared between underweight, normal-weight, and overweight individuals. The survival curve was constructed according to the 90-day mortality that was determined based on the different BMI groups. To identify the variables that were associated with mortality, we constructed a logistic regression model to analyze the variables with significant differences using the Kruskal-Wallis and chi-square tests. We analyzed the validation cohort to confirm the results.

The dependent variable of the logistic regression model was the 90-day mortality rate. The unadjusted model (Model 1) included intercept, age, sex, and different BMI groups (baseline: normal-weight). Models 2–5 were adjusted for the same variables used in Model 1. Model 2 further included the SOFA score on day 1, Model 3 further included comorbidities, Model 4 further included infection sites, and Model 5 further included glucose parameters. Model 6 included all the variables used in Models 1–5.

Finally, we analyzed the cytokine and biomarker data of 165 participants to investigate the biological mechanisms of sepsis-related mortality. We analyzed the biomarker differences across the three BMI groups in the low and high mNUTRIC score subgroups. The biomarkers that were analyzed included IL-6, IL-1RA, IL-10, IL-17, TNF-α, IFN-γ, and HLA-DR after ICU admission on days 1, 3, and 7.

## 3. Results

### 3.1. Baseline Characteristics of Patients with Low and High mNUTRIC Scores

The construction cohort of this study included 513 patients with sepsis who had high mNUTRIC scores and 285 patients with low mNUTRIC scores. Table 2 shows the baseline characteristics and outcomes of the study participants. The validation cohort of the study included 245 patients with sepsis who had high mNUTRIC scores and 245 patients with low mNUTRIC scores. Appendix A shows the baseline characteristics and outcomes of the study participants in the validation cohort.

In the construction cohort, the mean age of the participants was 67.1 years, and the patients with high mNUTRIC scores were older than those with low mNUTRIC scores (71.1 years vs. 60.0 years, respectively; *p* < 0.001). The mean ages were comparable in the validation cohort. Patients with sepsis who had high mNUTRIC scores had higher mean APACHE II scores (28.5 vs. 18.1; *p* < 0.001), Charlson comorbidity index (mean of 2.8 vs. 2.3; *p* < 0.001), and mean SOFA scores (10.2 vs. 6.8; *p* < 0.001). The APACHE II score, Charlson comorbidity index, and SOFA score were comparable in the validation cohort. Patients with sepsis who had high mNUTRIC scores also had a higher mean number of comorbidities (2.0 vs. 1.1; *p* < 0.001), such as coronary artery disease (32.0% vs. 14.0%; *p* < 0.001), stroke (23.2% vs. 9.8%; *p* < 0.001), hypertension (62.0% vs. 44.4%, *p* < 0.001), CKD (38.4% vs. 14.7%; *p* < 0.001), and diabetes mellitus (52.8% vs. 29.8%; *p* < 0.001), than those with low mNUTRIC scores. In the validation cohort, patients with sepsis who had high mNUTRIC scores also had a higher number of comorbidities; in addition, the prevalence values of stroke, hypertension, chronic kidney disease (CKD), and diabetes mellitus were comparable.

When comparing the high mNUTRIC and low mNUTRIC score groups, the proportions of patients with pneumonia (61.6% vs. 69.5%; *p* = 0.026) and soft tissue infection (3.5% vs. 7.9%; *p* = 0.009) were lower in the former than in the latter. First-day capillary glucose levels in patients admitted to the ICU were greater in the high mNUTRIC score group than in the low mNUTRIC score group (213.8% vs. 189.5%, respectively; *p* < 0.001). Moreover, the 7-day, 14-day, 28-day, and 90-day mortality rates were significantly lower in the low mNUTRIC score group than in the high mNUTRIC score group (7-day mortality rate: 16.2% vs. 6.3%, respectively; *p* < 0.001) (14-day mortality rate: 22.8% vs. 11.9%, respectively; *p* < 0.001) (28-day mortality rate: 32.0% vs. 21.8%, respectively; *p* < 0.001) (90-day mortality rate: 46.8% vs. 34.4%, respectively; *p* < 0.001).

### 3.2. Baseline Characteristics of Patients in the Different BMI Groups

The construction cohort of this study was divided into three groups: underweight (*n* = 149), normal-weight (*n* = 405), and overweight (*n* = 245) (Table 2). The validation cohort of the study was also divided into three groups according to BMI: underweight (*n* = 81), normal-weight (*n* = 241), and overweight (*n* = 170) (Appendix A).

In the construction cohort, the overweight group was the youngest (mean age = 64.7 years; *p* = 0.013), had the highest proportion of females (50.6%; *p* < 0.001), and had the highest mean SOFA score (9.7; *p* < 0.001) (the SOFA scores were comparable in the validation cohort). This group had the lowest proportions of patients with stroke (13.5%; *p* < 0.032), chronic obstructive pulmonary disease (COPD) (8.6%; *p* = 0.005), and cancer (16.0%; *p* = 0.002); highest proportion of patients with CKD (33.5%; *p* = 0.035), diabetes mellitus (53.9%; *p* < 0.001), bacteremia (11.4%; *p* = 0.016); and highest mean HbA1c (7.6%; *p* = 0.014), and capillary glucose (216.9 mg/dL; *p* < 0.001) levels. The underweight group was the oldest (mean age = 68.5 years; *p* < 0.001), had the lowest proportion of females, and had the lowest mean SOFA score (8.2; *p* < 0.001) and number of comorbidities (1.5; *p* = 0.014). This group had the lowest proportions of patients with coronary artery disease (17.4%; *p* = 0.023), CKD (21.5%, *p* = 0.035), and diabetes mellitus (32.1%; *p* < 0.001); highest proportions of patients with stroke (23.5%; *p* = 0.032), COPD (16.1%; *p* = 0.005), cancer (26.4%; *p* = 0.002), and pneumonia (79.9%; *p* < 0.001); longest ICU stay (14.0 days; *p* = 0.035); and lowest mean HbA1c (6.9%; *p* = 0.014) and capillary glucose (172.1 mg/dL; *p* < 0.001) levels. The proportion of patients with coronary artery disease, hypertension, and COPD and length of ICU stay were comparable in the validation cohort.

### 3.3. Survival Curve of Patients with Sepsis Classified According to the mNUTRIC Scores

Figure 1 shows that patients with sepsis who had low mNUTRIC scores had higher 90-day survival rates than those with high mNUTRIC scores in the construction cohort (63.2% ± 2.1% vs. 54.1% ± 1.7%, respectively; *p* = 0.002) and validation (73.8% ± 1.9% vs. 62.3% ± 2.2%; *p* < 0.001) cohort. Figure 2 shows the survival curves of patients with sepsis in the three BMI subgroups. In the construction cohort, underweight patients with lower mNUTRIC scores had a significantly higher 90-day survival rate (70.8% ± 4.2% vs. 54.2% ± 4.0%; *p* = 0.005), whereas overweight patients with lower mNUTRIC scores had a higher 90-day survival rate (66.2% ± 3.6% vs. 55.2% ± 3.2%; *p* = 0.059). Normal-weight patients did not show significantly different survival rates between the mNUTRIC score groups (58.3% ± 3.3% vs. 54.9% ± 2.3%; *p* = 0.309). In the validation cohort, only overweight patients with lower mNUTRIC scores had significantly higher 90-day survival rates (*p* < 0.001).

### 3.4. Survival Curves for Patients with Sepsis According to the Different BMI Groups

Figure 3 shows that there was no significant difference in the survival among the three BMI groups in the construction cohort (*p* = 0.372) and validation cohort (*p* = 0.132). Figure 4a shows that the 90-day survival rate of the underweight group with low mNUTRIC scores was significantly higher than that of the normal-weight group with low mNUTRIC scores (70.8% ± 4.2% vs. 58.3% ± 3.3%, respectively; *p* = 0.048) in the construction cohort. The survival rate was also higher in the overweight group than in the normal-weight group, although the difference was not significant (66.2% ± 3.6% vs. 58.3% ± 3.3%; *p* = 0.199). In the validation cohort, the 90-day survival rates were not significantly different among the different BMI groups with low mNUTRIC scores (Figure 4b).

Table 3 shows that the construction cohort group with underweight patients who had low mNUTRIC scores had the lowest proportion of females (underweight vs. normal-weight vs. overweight: 29.7% vs. 34.6% vs. 47.9%, respectively; *p* = 0.041), lowest SOFA scores on days 1, 3, and 7 (underweight vs. normal-weight vs. overweight: 6.1 ± 2.9 vs. 6.6 ± 3.1 vs. 7.6 ± 4.2, respectively, *p* = 0.039; 5.5 ± 2.2 vs. 6.5 ± 3.1 vs. 7.2 ± 3.5, respectively, *p* = 0.006; and 4.5 ± 2.3 vs. 6.4 ± 3.5 vs. 6.2 ± 3.4, respectively, *p* = 0.001), lowest proportion of patients with diabetes mellitus (underweight vs. normal-weight vs. overweight: 20.3% vs. 26.8% vs. 40.4%, respectively; *p* = 0.015), and highest proportion of patients with pneumonia (underweight vs. normal-weight vs. overweight: 78.1% vs. 74.0% vs. 54.4%, respectively; *p* = 0.007). The normal-weight group had the highest proportions of patients with stroke (underweight vs. normal-weight vs. overweight: 12.5% vs. 13.4% vs. 3.2%, respectively; *p* = 0.007) and cancer (underweight vs. normal-weight vs. overweight: 25% vs. 27.8% vs. 11.7%, *p* = 0.013), and the lowest mean HbA1c level (underweight vs. normal-weight vs. overweight: 7.0 ± 2.0 vs. 6.7 ± 1.7 vs. 7.7 ± 2.4, respectively; *p* = 0.027). In contrast, in the validation cohort group, there were higher proportions of patients with comorbidities, such as coronary artery disease and hypertension, in the overweight group with low mNUTRIC scores (Appendix A).

Table 4 shows the odds ratios of the logistic regression analysis of risk factors for 90-day mortality in patients with sepsis who had low mNUTRIC scores. In Model 1, the risk of mortality was reduced by nearly half (odds ratio (OR): 0.557; *p* = 0.082) in underweight patients. In Model 2, after further adjusting for the SOFA scores, similar trends were not observed for underweight patients. In Model 3, after further adjusting for comorbidities, the risk of mortality in underweight patients was still found to be reduced by nearly half (OR: 0.529; *p* = 0.075). In addition, Model 3 showed that the incidence of cancer significantly increased the risk of mortality by nearly four times (OR: 3.921; *p* < 0.001). In Model 4, after considering pneumonia, underweight patients still showed a risk of mortality that was decreased by nearly half (OR: 0.560; *p* = 0.085). In Model 5, after HbA1c levels were considered, the effect of being underweight was found to be insignificant. In contrast, HbA1c levels tended to decrease the risk of mortality by 21% (OR: 0.792; *p* = 0.078), although this difference was not significant. In Model 6, after considering all the above factors, only cancer was significantly associated with an increased risk of mortality (OR: 3.833; *p* = 0.026).

In patients with high mNUTRIC scores, there was no significant difference in 90-day mortality rates among the three BMI groups. Table 5 shows that the underweight group had the lowest proportion of females (underweight vs. normal-weight vs. overweight: 30.6% vs. 39.7% vs. 52.3%, respectively; *p* = 0.003), lowest mean SOFA score on day 1 (underweight vs. normal-weight vs. overweight: 9.8 ± 3.0 vs. 10.0 ± 3.5 vs. 11.0 ± 3.6, respectively; *p* = 0.007), lowest proportion of diabetes mellitus cases (underweight vs. normal-weight vs. overweight: 41.2% vs. 51.3% vs. 62.3%, respectively; *p* = 0.006), highest proportion of pneumonia cases (underweight vs. normal-weight vs. overweight: 81.2% vs. 63.5% vs. 47.0%, respectively; *p* < 0.001), and lowest mean capillary glucose level (underweight vs. normal-weight vs. overweight: 176.5 ± 74.8 vs. 216.2 ± 110.6 vs. 228.9 ± 108.1 mg/dL, respectively; *p* < 0.001).

### 3.5. Biomarkers of Patients with Sepsis

In the low mNUTRIC score subgroup, 53 patients underwent biomarker analysis (underweight: *n* = 10; normal-weight: *n* = 18; and overweight: *n* = 25). The normal-weight group had the lowest mean HLA-DR level on day 1 (underweight vs. normal-weight vs. overweight: 94.3 ± 8.5 vs. 82.1 ± 19.0 vs. 94.3 ± 6.0, respectively; *p* = 0.007) and day 3 (underweight vs. normal-weight vs. overweight: 91.8 ± 14.0 vs. 91.0 ± 5.0 vs. 93.2 ± 9.3, respectively; *p* = 0.047), and this was statistically significant. The normal-weight group also had the lowest levels of IL-6 on day 1 (underweight vs. normal-weight vs. overweight: 174.3 ± 253.4 vs. 83.7 ± 164.4 vs. 104.5 ± 281.4, respectively; *p* = 0.076) and day 3 (underweight vs. normal-weight vs. overweight: 188.1 ± 346.2 vs. 32.9 ± 47.7 vs. 129.7 ± 501.9, respectively; *p* = 0.072); IL-1RA on day 1 (underweight vs. normal-weight vs. overweight: 151.7 ± 123.4 vs. 49.5 ± 50.7 vs. 243.7 ± 394.9, respectively; *p* = 0.085) and day 3 (underweight vs. normal-weight vs. overweight: 147.3 ± 167.1 vs. 39.6 ± 49.8 vs. 77.6 ± 137.9, respectively; *p* = 0.061); and IFN-γ on day 1 (underweight vs. normal-weight vs. overweight: 55.0 ± 77.3 vs. 9.3 ± 11.3 vs. 12.5 ± 15.5, respectively; *p* = 0.068) (Table 6).

In the subgroup with patients who had high mNUTRIC scores, 112 patients underwent biomarker analysis (underweight: *n* = 20; normal-weight: *n* = 61; overweight: *n* = 31). The underweight group had the lowest mean HLA-DR level (underweight vs. normal-weight vs. overweight: 77.3 ± 17.1 vs. 84.7 ± 17.3 vs. 88.0 ± 13.4, respectively; *p* = 0.031) (Table 7).

## 4. Discussion

In this study, the 90-day survival curves showed that underweight patients with low mNUTRIC scores had significantly better survival rates than those with normal weights and low mNUTRIC scores (70.8% vs. 58.3%, respectively; *p* = 0.048). In the regression analysis, being underweight and the HbA1c levels predicted the survival of patients with sepsis who had low mNUTRIC scores. An increase of 1% of HbA1c had a trend to decrease 20% mortality in patients with sepsis and low mNUTRIC scores (ORR:0.792, *p* = 0.078). Both these characteristics were protective factors for survival. Cancer was an independent risk factor for survival, and patients with cancer had a nearly four-times higher risk of mortality (β_cancer_ = 3.921; *p* < 0.001) than those without cancer. Additionally, the mNUTRIC scores predicted 90-day survival rates in critically ill patients, but not in those with normal weights. This study revealed that when the mNUTRIC score of patients with sepsis was ≤5, being underweight was a protective factor and an indicator of survivability. We also highlighted the limitations of the mNUTRIC scores of patients with sepsis who had normal weights.

The present study revealed that being underweight was associated with better survival in patients with sepsis who had low mNUTRIC scores. Previous studies showed that being underweight may be a risk factor for patients with sepsis [18,19,20,21,22,24], while one study did not find a significant association between being underweight and outcomes in these patients [41]. Nevertheless, these studies did not consider the heterogeneity of patients with sepsis in the same BMI group. Most of them lacked baseline characteristics of clinical scores (NUTRIC score, SOFA score, and APACHE II) and laboratory data (HbA1C and blood glucose). In our analysis, the overall survival of patients with sepsis was not significantly different from that of those with normal weights. However, there were differences in the results of the subgroup analysis. Our results showed that better nutritional status of patients with sepsis was correlated with improved survivability in underweight patients. We also found that patients with low mNUTRIC scores had the lowest SOFA scores on days 1, 3, and 7 among the three BMI groups. The SOFA scores were used to evaluate the severity of sepsis and predict the outcomes of patients with sepsis [1]. In model 1 of Table 4, the underweight group had a trend to have better survival in patients with sepsis (0.557, *p* = 0.082). However, in adjusted model 2, the trend disappeared after we adjusted it by SOFA scores. Thus, the survival of the underweight patients might have been indicated by the low SOFA scores. Previous studies showed underweight patients had lower SOFA scores compared to normal weight and obesity patients [42,43]. However, the underweight group with lower SOFA scores did not have better survival in past studies. It may be related to heterogenecity of the underweight group. Unlike previous studies, we categorized patients with sepsis by mNUTRIC score which was used to differentiate nutrition status and severity. Our results revealed that underweight might not be a harmful factor when patients with sepsis have better nutrition status.

In addition, our analysis showed that cancer was an independent risk factor for sepsis-related mortality. Our previous study also showed that patients with active cancer had higher plasma IL-10 levels and mortality rates than those without cancer [3]. Normal-weight patients with low mNUTRIC scores had the highest cancer incidence among the three BMI groups (normal-weight vs. underweight vs. overweight: 25.0% vs. 27.8%, vs. 11.7%, respectively; *p* = 0.013), and there was no significant difference between the underweight and normal-weight patients (25.0% vs. 27.8%, respectively; *p* = 0.683). A previous study showed that overweight and obesity may contribute to the risk of cancer [44] and that cancer-related mortality may be associated with BMI changes after diagnosis [45]. Since our database lacked baseline BMI data before cancer, we could not determine whether normal-weight and underweight patients with cancer were overweight before losing weight due to cancer. However, a study by Flegal and colleagues revealed that being underweight significantly increased non-cancer-related mortality but was not associated with cancer-related mortality [20], suggesting that normal-weight patients had higher cancer mortality rates than underweight patients. This could also explain why normal-weight patients had higher mortality rates than underweight patients despite similar cancer rates in our study. In addition, higher HbA1c levels were associated with improved odds of patient survival. This finding was consistent with that of previous studies, which showed that diabetes may be associated with better outcomes [46].

In the validation cohort analysis, the 90-day survival curves did not show any significant difference in survival rates among the different BMI groups with low mNUTRIC scores. These findings were inconsistent with those of the construction cohort analysis. This is because the comorbidities differed between the construction and validation cohorts. The normal-weight group with low mNUTRIC scores in the construction cohort had higher cancer rates and stroke-associated mortality risk than the other BMI groups. In contrast, the overweight group with low mNUTRIC scores in the validation cohort had a higher cardiovascular-associated mortality risk than the other BMI groups. According to a study by Flegal and colleagues, obesity may increase the risk of cardiovascular-related mortality [20]. Therefore, the causes of mortality in the construction and validation cohorts might have been different. There were only 46 patients in the underweight group with low mNUTRIC scores. The sample size was too small to be statistically significant.

In patients with high mNUTRIC scores, mortality rates were not significantly different among the different BMI groups. Patients with high mNUTRIC scores had significantly higher mortality rates than those with low mNUTRIC scores. Nearly half of the patients with high mNUTRIC scores died within 90 days. Our analysis indicated that BMI did not affect survival when patients with sepsis had poor nutritional status during the critical phase. Moreover, the mNUTRIC scores predicted mortality in underweight and overweight patients. However, in the normal-weight subgroup analysis, the mortality rates were not significantly different between the high- and low-score groups, suggesting that the mNUTRIC score was not a good predictor of survival in normal-weight patients. In the validation cohort analysis, the normal-weight and overweight subgroup analyses were consistent with this finding.

Biomarker analysis further showed that patients in the normal-weight group with low mNUTRIC scores had the lowest HLA-DR levels (82.1%) among the three BMI groups. According to our previous study, HLA-DR levels of <87.2% on day 1 indicated immune dysfunction and predicted 28-day mortality [5]. In contrast, the mean HLA-DR level in patients with high mNUTRIC scores was 84.3%, which was lower than that on day 1. Thus, it was shown that patients with a high mortality risk had lower HLA-DR levels. Our study also revealed that IL-6, IL-1RA, and IFN-γ levels were the lowest in the normal-weight patients with low mNUTRIC scores. Previous studies have reported that the levels of cytokines such as IL-1, IL-6, IFN-γ, and TNF-α increase during sepsis [9,10,39,47] as an immune response to infection [10]. IL-1RA is a negative regulator of cytokine storms [39]. Normal-weight patients with low NUTRIC scores had low cytokine levels on day 1, suggesting a poor immune response against sepsis, which may lead to poor survival.

One strength of the present study was that it revealed that being underweight was an indicator of survivability in patients whose mNUTRIC scores were ≤5. Being underweight may not always be harmful if patients have optimal clinical nutritional status. Another strength was that we revealed the limitations of the mNUTRIC score for normal-weight patients. The development of novel or improved nutritional evaluation tools in the future is warranted to benefit normal-weight patients. In addition, we included both the construction cohort and validation cohort to confirm our findings and analyze the differences. Finally, we used biomarkers to validate our findings. This study validated our previous study on the HLA-DR level cut-off for predicting survival. To conclude, we used nutritional status to detect patients with high survival and analyzed biomarkers to explain the pathophysiological mechanisms.

Nevertheless, this study had some limitations. First, the study used body weights measured during ICU admission. BMI might be misclassified because of a lack of fluid balance adjustment [23]. Second, BMI does not represent complete body components, especially body fat. Previous studies have shown that obesity may reduce the risk of mortality via the secretion of leptin and soluble TNF receptor 2 during sepsis in adipose tissues [26]. Third, finger sugar, HbA1c, and biological data were not available for the validation cohort. Further studies are needed to clarify the relationship between complete body components and sepsis. In addition, the survival of patients with sepsis may be affected by multiple factors, such as age, sex, comorbidity, infection sites, and antibiotics. Although we used controls for age, sex, multiple comorbidities, and infection sites, we could not make a direct causal correlation between nutritional status and survival, which would require further evidence. Since this was a single-center retrospective study, we need further prospective trials to confirm our results.

## 5. Conclusions

The major contribution of our study was that it revealed that being underweight may not always be harmful if patients have optimal clinical nutritional status. Our findings have important implications for clinicians, as they can guide the application of therapeutic strategies and provide a better understanding of the underlying mechanisms of sepsis-related mortality.

## Figures and Tables

**Figure 1 nutrients-13-01873-f001:**
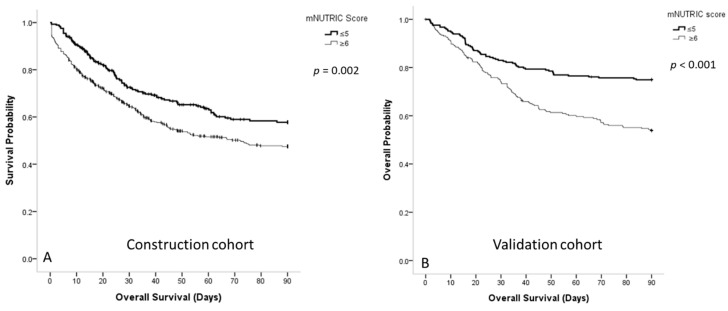
(**A**) Survival curve for the patients with sepsis in the medical intensive care unit according to the mNUTRIC scores in the construction cohort; (**B**) survival curve for the patients with sepsis in the medical intensive care unit according to the mNUTRIC scores in the validation cohort.

**Figure 2 nutrients-13-01873-f002:**
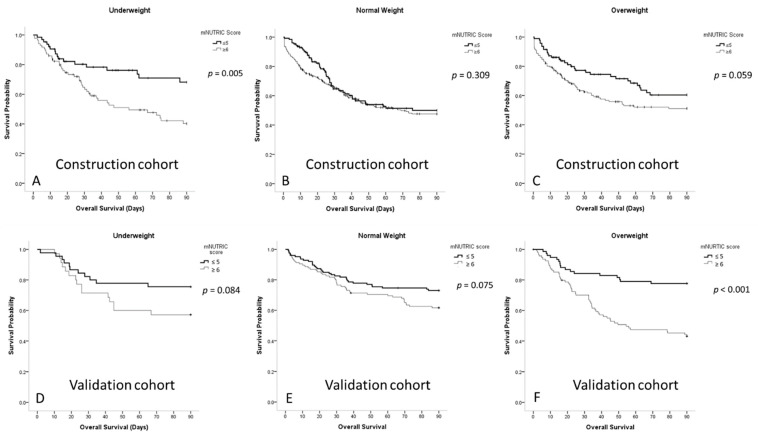
(**A**) Survival curves for underweight patients with sepsis in the medical intensive care unit according to the mNUTRIC scores in the construction cohort; (**B**) survival curves for normal-weight patients with sepsis in the medical intensive care unit according to the mNUTRIC scores in the construction cohort; (**C**) survival curves for overweight patients with sepsis in the medical intensive care unit according to the mNUTRIC scores in the construction cohort; (**D**) survival curves for underweight patients with sepsis in the medical intensive care unit according to the mNUTRIC scores in the validation cohort; (**E**) survival curves for normal-weight patients with sepsis in the medical intensive care unit according to the mNUTRIC scores in the validation cohort; (**F**) survival curves for overweight patients with sepsis in the medical intensive care unit according to the mNUTRIC scores in the validation cohort.

**Figure 3 nutrients-13-01873-f003:**
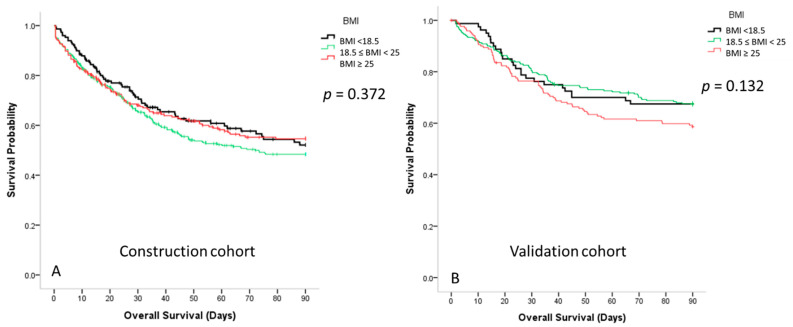
(**A**) Survival curves for the patients with sepsis in the medical intensive care unit according to the body mass index (BMI) in the construction cohort; (**B**) survival curves for the patients with sepsis in the medical intensive care unit according to the body mass index in the validation cohort.

**Figure 4 nutrients-13-01873-f004:**
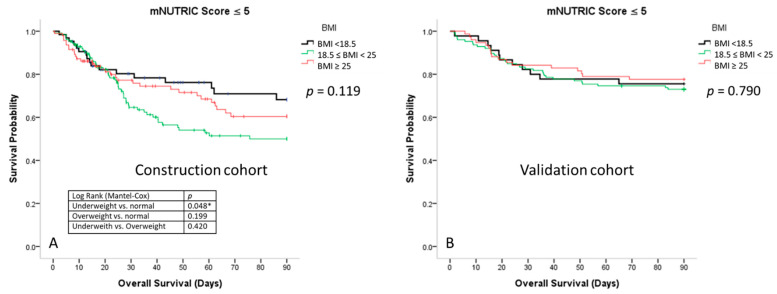
(**A**) Survival curves for the patients with sepsis in the medical intensive care unit who had low mNUTRIC scores (≤5) according to body mass index in the construction cohort; (**B**) survival curves for the patients with sepsis in the medical intensive care unit who had low mNUTRIC scores (≤5) according to body mass index in the validation cohort. * The 90-day survival rate of the underweight group with low mNUTRIC scores was significantly higher than that of the normal-weight group with low mNUTRIC scores (70.8% ± 4.2% vs. 58.3% ± 3.3%, respectively; *p* = 0.048).

**Table 1 nutrients-13-01873-t001:** modified NUTRIC score.

Variable	Range	Points
Age	<50	0
	≥50 and <75	1
	≥75	2
APACHE II	<15	0
	≥15 and <20	1
	≥20 and <28	2
	≥28	3
SOFA	<6	0
	≥6 and <10	1
	≥10	2
Number of comorbidities	0–1	0
	≥2	1
Days from hospital to ICU admission	≥0 and <1	0
	≥1	1

APACHE II: acute physiology and chronic health evaluation II; SOFA: sequential organ failure assessment; ICU: Intensive Care Unit.

**Table 2 nutrients-13-01873-t002:** Baseline characteristics and outcomes of the patients with sepsis in the construction cohort of the study.

mNUTRIC Score on the First Day	All (*n* = 799)	mNUTRIC Score ≥ 6 (*n* = 513)	mNUTRIC Score ≤ 5 (*n* = 286)	*p*	Under-Weight (*n* = 149)	Normal-Weight (*n* = 405)	Overweight (*n* = 245)	*p*
Demographic characteristics, mean ± SD								
Age (years)	67.1 ± 14.9	71.1 ± 12.5	60.0 ± 16.1	<0.001 ***	68.5 ± 15.2	68.0 ± 14.4	64.7 ± 15.2	0.013 *
BMI, kg/m^2^	22.8 ± 4.9	22.9 ± 4.7	22.7 ± 5.2	0.464	16.5 ± 1.5	21.7 ± 1.8	28.8 ± 3.5	<0.001 ***
Sex (female), *n* (%)	323 (40.5)	215 (41.9)	108 (37.9)	0.268	45 (30.2)	154 (38.0)	124 (50.6)	<0.001 ***
Score, mean ± SD								
APACHE II	24.7 ± 8.4	28.5 ± 6.9	18.1 ± 6.4	<0.001 ***	23.8 ± 7.5	25.3 ± 8.3	24.4 ± 9.0	0.094
Charlson comorbidity index	2.6 ± 2.0	2.8 ± 1.9	2.3 ± 2.1	<0.001 ***	2.7 ± 2.1	2.7 ± 2.0	2.4 ± 1.9	0.315
SOFA	9.0 ± 3.8	10.2 ± 3.4	6.8 ± 3.5	<0.001 ***	8.2 ± 3.5	8.9 ± 3.7	9.7 ± 4.1	0.001 **
mNUTRIC	6.0 ± 1.8	7.1 ± 1.0	4.1 ± 1.1	<0.001 ***	5.8 ± 1.8	6.1 ± 1.7	5.9 ± 1.8	0.127
Comorbidities (number)	1.7 ± 1.2	2.0 ± 1.1	1.1 ± 1.0	<0.001 ***	1.5 ± 1.2	1.8 ± 1.2	1.6 ± 1.2	0.014 *
Comorbidities, *n* (%)								
Coronary artery disease	204 (25.5)	164 (32.0)	40 (14.0)	<0.001 ***	26 (17.4)	117 (28.9)	61 (24.9)	0.023 *
History of stroke	147 (18.4)	119 (23.2)	28 (9.8)	<0.001 ***	35 (23.5)	79 (19.5)	33 (13.5)	0.032 *
Hypertension	444 (55.6)	318 (62.0)	126 (44.4)	<0.001 ***	73 (49.0)	225 (55.7)	146 (59.6)	0.121
COPD	117 (14.6)	82 (16.0)	34 (11.9)	0.119	24 (16.1)	72 (17.8)	21 (8.6)	0.005 **
Cancer	190 (23.8)	127 (25.0)	62 (21.8)	0.309	39 (26.4)	112 (28.0)	39 (16.0)	0.002 **
CKD	239 (29.9)	197 (38.4)	42 (14.7)	<0.001 ***	32 (21.5)	125 (30.9)	82 (33.5)	0.035 *
Liver cirrhosis	69 (8.6)	50 (9.7)	19 (6.7)	0.138	9 (6.0)	33 (8.1)	27 (11.0)	0.206
Diabetes mellitus	356 (44.6)	271 (52.8)	85 (29.8)	<0.001 ***	48 (32.1)	176 (43.5)	132 (53.9)	<0.001 ***
Site of suspected infection, *n* (%)								
Lung	515 (64.5)	316 (61.6)	198 (69.5)	0.026 *	119 (79.9)	271 (66.9)	125 (51.0)	<0.001 ***
UTI	169 (21.2)	113 (22.0)	56 (19.6)	0.431	13 (22.1)	79 (19.5)	57 (23.3)	0.496
Intraabdominal infection	57 (7.1)	40 (7.8)	17 (6.0)	0.336	5 (3.4)	33 (8.1)	19 (7.8)	0.137
Soft tissue infection	40 (5.0)	18 (3.5)	22 (7.7)	0.009 **	8 (5.4)	14 (3.5)	18 (7.3)	0.086 ^†^
Bacteraemia	61 (7.6)	44 (8.6)	17 (6.0)	0.183	6 (4.0)	27 (6.7)	28 (11.4)	0.016 *
Baseline glucose and HbA1c, mean ± SD								
HbA1c (%) **	7.3 ± 2.2	7.3 ± 2.3	7.2 ± 2.1	0.453	6.9 ± 2.0	7.1 ± 2.1	7.6 ± 2.4	0.014 *
Glucose (mg/dL)	205.7 ± 117.6	213.8 ± 106.2	189.5 ± 136.7	<0.001 ***	172.1 ± 73.7	210.7 ± 135.9	216.9 ± 102.2	<0.001 ***
Admission days, mean ± SD								
ICU days	12.5 ± 9.7	12.3 ± 9.5	12.9 ± 10.1	0.411	14.0 ± 10.0	12.0 ± 9.2	12.3 ± 10.3	0.035 *
LOS	30.1 ± 28.4	30.5 ± 31.0	29.5 ± 23.0	0.364	35.6 ± 32.3	28.6 ± 28.1	29.3 ± 26.1	0.070 ^†^
Mortality, *n* (%)								
7-day mortality	102 (12.8)	83 (16.2)	18 (6.3)	<0.001 ***	13 (8.7)	54 (13.3)	35 (14.3)	0.245
14-day mortality	152 (19)	117 (22.8)	34 (11.9)	<0.001 ***	23 (15.4)	81 (20.0)	48 (19.6)	0.461
28-day mortality	227 (28.4)	164 (32.0)	62 (21.8)	0.002 **	35 (23.5)	120 (29.6)	72 (29.4)	0.335
90-day mortality	339 (42.4)	240 (46.8)	98 (34.4)	0.001 **	60 (40.3)	181 (44.7)	98 (40.0)	0.422

*** *p* < 0.001; ** *p* < 0.01; * *p* < 0.05; ^†^ *p* < 0.09; abbreviations: SD, standard deviation; mNUTRIC, modified nutrition risk in critically ill; BMI, body mass index; APACHE, acute physiology and chronic health evaluation; SOFA, sequential organ failure assessment; COPD, chronic obstructive pulmonary disease; CKD, chronic kidney disease; UTI, urinary tract infection; HbA1c, hemoglobin A1c; ICU, intensive care unit; LOS, length of stay.

**Table 3 nutrients-13-01873-t003:** Baseline characteristics and outcomes of patients with sepsis who had low modified NUTRIC scores.

	mNUTRIC Score ≤ 5 (*n* = 286)	Underweight (*n* = 64)	Normal-Weight (*n* = 128)	Overweight (*n* = 94)	*p*
Demographic characteristics, mean ± SD					
Age (years)	60.0 ± 16.1	61.8 ± 15.4	60.4 ± 15.9	58.2 ± 16.7	0.283
BMI, kg/m^2^	22.7 ± 5.2	16.4 ± 1.6	21.5 ± 1.9	28.6 ± 3.5	<0.001 ***
Sex (female), *n* (%)	108 (37.9)	19 (29.7)	44 (34.6)	45 (47.9)	0.041 *
Score, mean ± SD					
APACHE II	18.1 ± 6.4	18.8 ± 6.4	18.1 ± 6.5	17.4 ± 6.3	0.276
Charlson comorbidity index	2.3 ± 2.1	2.6 ± 2.3	2.5 ± 2.3	2.0 ± 1.9	0.354
SOFA on day 1	6.8 ± 3.5	6.1 ± 2.9	6.6 ± 3.1	7.6 ± 4.2	0.039 *
SOFA on day 3	6.5 ± 3.1	5.5 ± 2.2	6.5 ± 3.1	7.2 ± 3.5	0.006 **
SOFA on day 7	5.9 ± 3.3	4.5 ± 2.3	6.4 ± 3.5	6.2 ± 3.4	0.001 **
mNUTRIC	4.1 ± 1.1	4.1 ± 1.1	4.1 ± 1.0	4.0 ± 1.1	0.550
Comorbidities (number)	1.1 ± 1.0	1.0 ± 0.9	1.2 ± 1.1	1.0 ± 1.1	0.112
Comorbidities, *n* (%)					
Coronary artery disease	40 (14.0)	7 (10.9)	19 (15)	14 (14.9)	0.720
History of stroke	28 (9.8)	8 (12.5)	17 (13.4)	3 (3.2)	0.030 *
Hypertension	126 (44.4)	23 (35.9)	57 (45.2)	46 (48.9)	0.262
COPD	34 (11.9)	7 (10.9)	23 (18.1)	4 (4.3)	0.007 **
Cancer	62 (21.8)	16 (25.0)	35 (27.8)	11 (11.7)	0.013 *
CKD	42 (14.7)	5 (7.8)	21 (16.5)	16 (17.0)	0.206
Liver cirrhosis	19 (6.7)	4 (6.3)	7 (5.5)	8 (8.5)	0.669
Diabetes mellitus	85 (29.8)	13 (20.3)	34 (26.8)	38 (40.4)	0.015 *
Site of suspected infection, *n* (%)					
Lung	198 (69.5)	50 (78.1)	94 (74.0)	54 (54.4)	0.007 **
UTI	56 (19.6)	12 (18.8)	24 (18.9)	20 (21.3)	0.889
Intraabdominal infection	17 (6.0)	1 (1.6)	12 (9.4)	4 (4.3)	0.066 ^†^
Soft tissue infection	22 (7.7)	5 (7.8)	7 (5.5)	10 (10.6)	0369
Bacteraemia	17 (6.0)	3 (4.7)	7 (5.5)	7 (7.4)	0.741
Baseline glucose and HbA1c, mean ± SD					
HbA1c (%) **	7.2 ± 2.1	7.0 ± 2.0	6.7 ± 1.7	7.7 ± 2.4	0.027 *
Glucose (mg/dL)	189.5 ± 136.7	165.8 ± 72.4	197.8 ± 184.2	194.5 ± 86.6	0.069 ^†^
Admission days, mean ± SD					
ICU days	12.9 ± 10.1	14.3 ± 11.2	12.3 ± 8.2	12.9 ± 11.7	0.399
LOS	29.5 ± 23.0	32.7 ± 27.3	26.5 ± 18.3	31.5 ± 25.2	0.630
Mortality, *n* (%)					
7-day mortality	18 (6.3)	3 (4.7)	7 (5.5)	8 (8.5)	0.551
14-day mortality	34 (11.9)	8 (12.5)	13 (10.5)	13 (13.5)	0.708
28-day mortality	62 (21.8)	12 (18.8)	30 (23.6)	20 (21.3)	0.736
90-day mortality	98 (34.4)	17 (26.6)	50 (39.4)	31 (33.0)	0.200 ^a^

*** *p* < 0.0001; ** *p* < 0.01; * *p* < 0.05; ^†^ *p* < 0.09; ^a^ underweight vs. normal-weight, *p* = 0.048. Abbreviations: SD, standard deviation; mNUTRIC, modified nutrition risk in critically ill; BMI, body mass index; APACHE, acute physiology and chronic health evaluation; SOFA, sequential organ failure assessment; COPD, chronic obstructive pulmonary disease; CKD, chronic kidney disease; UTI, urinary tract infection; HbA1c, hemoglobin A1c; ICU, intensive care unit; LOS, length of stay.

**Table 4 nutrients-13-01873-t004:** Results (odds ratios) of the logistic regression analysis of risk factors for 90-day mortality in patients with sepsis who had low modified NUTRIC scores.

	Model 1	Model 2	Model 3	Model 4	Model 5	Model 6
Intercept	0.598	0.330	0.437	0.651	1.578	1.433
Underweight	0.557 ^†^	0.571	0.529 ^†^	0.560 ^†^	0.577	0.594
Overweight	0.756	0.712	0.926	0.735	0.853	0.905
Age	1.002	1.005	1.002	1.003	1.011	1.012
Sex (female), *n* (%)	0.950	0.924	0.936	0.951	0.661	0.544
Score						
SOFA on day 1		1.062				1.002
Comorbidities						
Coronary artery disease			0.785			0.593
COPD			0.948			1.927
Cancer			3.921 ***			3.833 *
DM			0.821			0.758
Site of suspected infection						
Lung				0.841		0.717
Baseline glucose and HbA1c						
HbA1c (%)					0.792 ^†^	0.819
Goodness-of-fit						
*p*	0.498	0.315	0.001 **	0.586	0.206	0.242
r^2^	0.012	0.021	0.086	0.013	0.054	0.10

*** *p* < 0.001; ** *p* < 0.01; * *p* < 0.05; ^†^ *p* < 0.09; abbreviations: mNUTRIC, modified nutrition risk in critically ill; SOFA, sequential organ failure assessment; COPD, chronic obstructive pulmonary disease; DM, diabetes mellitus; HbA1c, hemoglobin A1c.

**Table 5 nutrients-13-01873-t005:** Baseline characteristics and outcomes of patients with sepsis who had high modified NUTRIC scores.

	mNUTRIC Score ≥ 6 (*n* = 513)	Underweight (*n* = 85)	Normal-Weight (*n* = 277)	Overweight (*n* = 151)	*p*
Demographic characteristics, mean ± SD					
Age (years)	71.1 ± 12.5	73.6 ± 12.9	71.6 ± 12.2	68.8 ± 12.5	0.006 **
BMI, kg/m^2^	22.9 ± 4.7	16.5 ± 1.5	21.8 ± 1.8	28.5 ± 3.5	<0.001 ***
Sex (female), *n* (%)	215 (41.9)	26 (30.6)	110 (39.7)	79 (52.3)	0.003 **
Score, mean ± SD					
APACHE II	24.7 ± 8.4	27.4 ± 5.9	28.6 ± 6.8	28.8 ± 7.5	0.445
Charlson comorbidity index	2.6 ± 2.0	2.7 ± 2.0	2.8 ± 1.9	2.6 ± 1.8	0.633
SOFA on day 1	9.0 ± 3.8	9.8 ± 3.0	10.0 ± 3.5	11.0 ± 3.6	0.007 **
SOFA on day 3	8.4 ± 3.6	7.9 ± 3.0	8.3 ± 3.8	8.8 ± 3.6	0.118
SOFA on day 7	7.3 ± 3.8	6.9 ± 3.0	7.2 ± 3.9	7.9 ± 4.2	0.346
mNUTRIC	7.1 ± 1.0	7.1 ± 0.9	7.1 ± 1.0	7.1 ± 1.0	0.796
Comorbidities (number)	2.0 ± 1.1	1.9 ± 1.2	2.0 ± 1.1	2.0 ± 1.1	0.272
Comorbidities, *n* (%)					
Coronary artery disease	164 (32.0)	19 (22.4)	98 (35.4)	47 (31.1)	0.327
History of stroke	119 (23.2)	27 (31.8)	62 (22.4)	30 (19.9)	0.103
Hypertension	318 (62.0)	50 (58.8)	168 (60.6)	100 (66.2)	0.423
COPD	82 (16.0)	17 (20.2)	48 (17.3)	17 (11.3)	0.142
Cancer	127 (25.0)	23 (27.4)	76 (27.8)	28 (17.7)	0.099
CKD	197 (38.4)	27 (31.8)	104 (37.5)	66 (43.7)	0.177
Liver cirrhosis	50 (9.7)	5 (5.9)	26 (9.4)	19 (12.6)	0.239
Diabetes mellitus	271 (52.8)	35 (41.2)	142 (51.3)	94 (62.3)	0.006 **
Site of suspected infection, *n* (%)					
Lung	316 (61.6)	69 (81.2)	176 (63.5)	71 (47.0)	<0.001 ***
UTI	113 (22.0)	21 (24.7)	55 (19.9)	37 (24.5)	0.437
Intraabdominal infection	40 (7.8)	4 (4.7)	21 (7.6)	15 (9.9)	0.349
Soft tissue infection	18 (3.5)	3 (3.5)	7 (2.5)	8 (5.3)	0.330
Bacteremia	44 (8.6)	3 (3.5)	20 (7.2)	21 (13.9)	0.012 *
Baseline glucose and HbA1c, mean ± SD					
HbA1c (%) **	7.3 ± 2.3	6.7 ± 2.1	7.3 ± 2.2	7.6 ± 2.5	0.078 ^†^
Glucose (mg/dL)	213.8 ± 106.2	176.5 ± 74.8	216.2 ± 110.6	228.9 ± 108.1	0.001 **
Admission days, mean ± SD					
ICU days	12.3 ± 9.5	13.8 ± 9.0	12.0 ± 9.6	11.9 ± 9.4	0.083 ^†^
LOS	30.5 ± 31.0	37.7 ± 35.6	29.7 ± 31.6	28.0 ± 26.6	0.077 ^†^
Mortality, *n* (%)					
7-day mortality	83 (16.2)	10 (11.8)	46 (16.6)	27 (17.9)	0.454
14-day mortality	117 (22.8)	15 (17.6)	67 (24.2)	35 (23.2)	0.450
28-day mortality	164 (32.0)	23 (27.1)	89 (32.1)	52 (34.4)	0.504
90-day mortality	240 (46.8)	43 (50.6)	130 (46.9)	67 (44.4)	0.654

*** *p* < 0.001; ** *p* < 0.01; * *p* < 0.05, ^†^ *p* < 0.09; abbreviations: SD, standard deviation; mNUTRIC, modified nutrition risk in critically ill; BMI, body mass index; APACHE, acute physiology and chronic health evaluation; SOFA, sequential organ failure assessment; COPD, chronic obstructive pulmonary disease; CKD, chronic kidney disease; UTI, urinary tract infection; HbA1c, hemoglobin A1c; ICU, intensive care unit; LOS, length of stay.

**Table 6 nutrients-13-01873-t006:** Biomarkers of patients with sepsis who had low modified NUTRIC scores.

	mNUTRIC Score ≤ 5 (*n* = 53)	Underweight (*n* = 10)	Normal-Weight (*n* = 18)	Overweight (*n* = 25)	*p*
IL-6 on day 1 (10/18/25)	110.6 ± 239.9	174.3 ± 253.4	83.7 ± 164.4	104.5 ± 281.4	0.076 ^†^
IL-6 on day 3 (10/17/24)	108.8 ± 376.3	188.1 ± 346.2	32.9 ± 47.7	129.7 ± 501.9	0.072 ^†^
IL-6 on day 7 (10/15/21)	75.5 ± 241.4	51.5 ± 58.4	75.4 ± 155.9	87.0 ± 344.9	0.274
IL-1RA on day 1	160.4 ± 288.2	151.7 ± 123.4	49.5 ± 50.7	243.7 ± 394.9	0.085 ^†^
IL-1RA on day 3	78.6 ± 126.6	147.3 ± 167.1	39.6 ± 49.8	77.6 ± 137.9	0.061 ^†^
IL-1RA on day 7	84.9 ± 142.2	108.5 ± 133.1	104.4 ± 166.2	58.9 ± 130.3	0.351
IL-10 on day 1	30.3 ± 41.4	41.0 ± 53.2	21.7 ± 24.6	32.3 ± 46.0	0.826
IL-10 on day 3	35.7 ± 36.8	36.3 ± 51.2	17.3 ± 18.6	27.1 ± 39.7	0.784
IL-10 on day 7	32.0 ± 74.9	40.3 ± 81.9	42.9 ± 109.0	20.3 ± 32.0	0.792
IL-17 on day 1	10.6 ± 17.8	19.6 ± 29.3	6.9 ± 14.0	9.7 ± 13.6	0.124
IL-17 on day 3	12.5 ± 19.5	23.7 ± 30.4	7.8 ± 13.4	11.1 ± 16.3	0.099
IL-17 on day 7	9.5 ± 13.4	8.9 ± 8.1	9.8 ± 18.9	9.6 ± 11.0	0.901
TNF-α on day 1	33.5 ± 31.8	45.0 ± 29.8	20.7 ± 11.7	38.1 ± 39.4	0.159
TNF-α on day 3	29.8 ± 23.7	34.8 ± 26.8	30.4 ± 20.7	27.2 ± 25	0.276
TNF-α on day 7	23.1 ± 15.6	22.7 ± 18.6	22.9 ± 15.0	23.4 ± 15.3	0.963
IFN-R on day 1	19.4 ± 38.6	55.0 ± 77.3	9.3 ± 11.3	12.5 ± 15.5	0.068 ^†^
IFN-R on day 3	23.1 ± 43.7	64.1 ± 83.9	12.2 ± 13.2	13.7 ± 19.2	0.600
IFN-R on day 7	21.0 ± 34.6	42.1 ± 60.8	16.1 ± 17.6	14.4 ± 23.0	0.677
HLA-DR on day 1	90.1 ± 13.5	94.3 ± 8.5	82.1 ± 19.0	94.3 ± 6.0	0.007 **
HLA-DR on day 3	92.2 ± 9.3	91.8 ± 14.0	91.0 ± 5.0	93.2 ± 9.3	0.047 *
HLA-DR on day 7	93.3 ± 7.7	95.0 ± 3.8	90.0 ± 8.5	94.0 ± 8.4	0.361

** *p* < 0.01; * *p* < 0.05; ^†^ *p* < 0.09; abbreviations: mNUTRIC, modified nutrition risk in critically ill; IL-6, interleukin 6; IL-1RA, interleukin 1 receptor antagonist; IFN-R, interferon receptor; IL-10, interleukin 10; IL-17, interleukin 17; TNF, tumor necrosis factor; HLA-DR, human leukocyte antigen receptor.

**Table 7 nutrients-13-01873-t007:** Biomarkers of patients with sepsis who had high modified NUTRIC scores.

	mNUTRIC Score ≥ 6 (*n* = 112)	Underweight (*n* = 20)	Normal-Weight (*n* = 61)	Overweight (*n* = 31)	*p*
IL-6 on day 1 (20/61/31)	623.2 ± 2281.8	926.2 ± 3544.1	354.8 ± 1061.3	955.8 ± 2955.5	0.621
IL-6 on day 3 (20/54/26)	109.9 ± 320.5	61.8 ± 65.1	106.1 ± 200.1	154.8 ± 545.0	0.831
IL6 on day 7 (18/46/23)	50.1 ± 106.1	71.0 ± 195.9	40.5 ± 42.3	52.7 ± 100.8	0.875
IL-1RA on day 1	854.7 ± 2526.7	548.1 ± 1285.0	983.5 ± 3087.2	799.1 ± 1847.9	0.854
IL-1RA on day 3	361.7 ± 1474.9	748.2 ± 2951.2	279.3 ± 832.1	235.4 ± 613.7	0.936
IL-1RA on day 7	302.0 ± 1648.7	912.9 ± 3608.9	108.2 ± 165.1	211.6 ± 366.4	0.504
IL-10 on day 1	177.5 ± 446.8	130.2 ± 256.6	170.1 ± 448.4	222.5 ± 539.5	0.298
IL-10 on day 3	72.2 ± 222.2	47.5 ± 57.1	98.8 ± 297.1	36.0 ± 49.2	0.549
IL-10 on day 7	47.8 ± 168.7	45.9 ± 73.1	55.2 ± 225.8	34.5 ± 51.0	0.705
IL-17 on day 1	16.5 ± 34.2	29.4 ± 56.2	12.0 ± 18.2	17.1 ± 38.3	0.166
IL-17 on day 3	18.9 ± 40.0	40.1 ± 71.1	11.4 ± 22.0	18.3 ± 31.6	0.247
IL-17 on day 7	24.8 ± 98.7	66.2 ± 211.4	14.2 ± 25.3	13.7 ± 21.3	0.599
TNF-α on day 1	83.0 ± 143.3	53.8 ± 35.3	79.3 ± 108.1	109.3 ± 224.7	0.775
TNF-α on day 3	58.2 ± 87.0	35.6 ± 24.3	58.5 ± 87.5	74.6 ± 112.2	0.351
TNF-α on day 7	44.9 ± 38.3	37.5 ± 25.8	41.8 ± 30.6	56.8 ± 55.6	0.946
IFN-R on day 1	36.8 ± 88.9	64.1 ± 147.6	29.0 ± 72.4	34.4 ± 65.8	0.540
IFN-R on day 3	40.7 ± 142.9	122.9 ± 299.7	21.0 ± 47.3	18.4 ± 33.2	0.256
IFN-R on day 7	70.1 ± 389.5	233.3 ± 839.4	36.1 ± 100.0	10.1 ± 17.0	0.062 ^†^
HLA-DR on day 1	84.3 ± 16.5	77.3 ± 17.1	84.7 ± 17.3	88.0 ± 13.4	0.031 *
HLA-DR on day 3	86.5 ± 15.4	78.1 ± 20.2	87.4 ± 14.7	90.7 ± 10.0	0.060 ^†^
HLA-DR on day 7	92.8 ± 11.3	89.7 ± 14.5	92.0 ± 12.3	96.7 ± 3.3	0.218

* *p* < 0.05; ^†^ *p* < 0.09; abbreviations: mNUTRIC, modified nutrition risk in critically ill; IL-6, interleukin 6; IL-1RA, interleukin-1 receptor antagonist; IFN-R, interferon receptor; IL-10, interleukin 10; IL-17, interleukin 17; TNF, tumor necrosis factor; HLA-DR, human leukocyte antigen receptor.

## Data Availability

The data presented in this study are available on request from the corresponding author. The data are not publicly available due to patients’ privacy.

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
