# Peer review of "Impact of Body Mass Index on the Survival of Patients with Sepsis with Different Modified NUTRIC Scores"

_nutrients, 2021, doi:10.3390/nu13061873_

Round 1

Reviewer 1 Report

Congratulations on a very interesting paper.

Author Response

Many thanks for your valuable comment and encouragement. 

Reviewer 2 Report

I would like to congratulate the authors for their nice and thorough analysis. In addition, the aim of searching specific phenotypes in sepsis that could improve the outcomes is undoubtedly desirable.

Globally comments: 

These authors have retrospectively analyzed patients diagnosed with sepsis
at a single hospital. Their goal was to identify if nutritional phenotype influences survival. There are many results and the stream
of information is sometimes difficult to digest. Moreover, I have missed a more detailed discussion about their main finding: why underweighted patients with low mNUTRIC scores have better survival?

Specific comments:

Abstract: The retrospective nature of the study should be clearly stated.

Methods:

Page 3. Line 110: A figure explaining the modified NUTRIC score would be of interest.

Page 3. Line 115-116: The authors have chosen 6 as their cut-off point in NUTRIC score based on the results of their sample. Is this congruent with previous literature?

Page 3. Line 125: Biomarkers data were collected from day 1 after ICU admission. It would have been preferable to collect first sample on ICU admission to have baseline values. If this has not been possible, the authors should discuss on this shortcoming.

Results:

Page 7. Line 255: The reference to figure 4b seems to be wrong. The authors refer in the text to the different 90-day survival rate between BMI groups when in line 248 the authors state the opposite. In addition, figure 4b refers to patients who had low mNUTRIC scores according to BMI.

Discussion:

Page 14. Line 358: When referring to HbA1c levels, the authors should comment on which levels, low or high HbA1c levels predict the survival of patients with sepsis and low mNUTRIC scores?

Page 16. Line 447: When discussing limitations, the authors should declare the retrospective nature of their design as one of the most important limitations of their study, and the need of further prospective trials to confirm their results. In addition, it is a single center study in which cut-off point regarding mNUTRIC score is obtained from your sample, that is external validity is likely low.

Conclusions:

Page 16. Lines 460-1: Authors should be careful in stating their conclusions. First, it is a retrospective study and second, as authors have correctly stated in their limitations, the survival of patients with sepsis may be influenced by multiple factors. Therefore, stating that their study reveals that being underweight was an indicator of survival in patients with sepsis who had low mNUTRIC scores <5 could be a leap of faith. Please make sure you synchronize your statements with your findings taking in account shortcomings. 

Author Response

Many Thanks for your valuable comments and suggestions which have led to significant improvement on the presentation and quality of this paper. In what follows, we shall detail the changes we have made on the paper.

Globally comments:

These authors have retrospectively analyzed patients diagnosed with sepsis at a single hospital. Their goal was to identify if nutritional phenotype influences survival. There are many results and the stream of information is sometimes difficult to digest. Moreover, I have missed a more detailed discussion about their main finding: why underweight patients with low mNUTRIC scores have better survival?

A: Thanks for your valuable comments.

We have revised the discussion paragraph and added references to support the main finding. (page 15, paragraph 2)

The present study revealed that being underweight was associated with better survival in patients with sepsis who had low mNUTRIC scores. Previous studies showed that being underweight may be a risk factor for patients with sepsis [18–22, 24], while one study did not find a significant association between being underweight and outcomes in these patients [41]. Nevertheless, these studies did not consider the heterogeneity of patients with sepsis in the same BMI group. Most of them lacked baseline characteristics of clinical scores (NUTRIC score, SOFA score, and APACHE II) and laboratory data (HbA1C and blood glucose). In our analysis, the overall survival of patients with sepsis was not significantly different from that of those with normal weights. However, there were differences in the results of the subgroup analysis. Our results showed that better nutritional status of patients with sepsis was correlated with improved survivability in underweight patients. We also found that patients with low mNUTRIC scores had the lowest SOFA scores on days 1, 3, and 7 among the three BMI groups.The SOFA scores were used to evaluate the severity of sepsis and predict the outcomes of patients with sepsis [1]. In model 1 of Table 4, the underweight group had a trend to have better survival in patients with sepsis (0.557, p=0.082). However, in adjusted model 2, the trend disappeared after we adjusted it by SOFA scores. Thus, the survival of the underweight patients might have been indicated by the low SOFA scores. Previous studies showed underweight patients had lower SOFA scores compared to normal weight and obesity patients [42,43]. However, the underweight group with lower SOFA scores didn’t have better survival in past studies. It may be related to heterogenecity of the underweight group. Unlike previous studies, we categorized patients with sepsis by mNUTRIC score which was used to differentiate nutrition status and severity. Our results revealed that underweight might not be a harmful factor when patients with sepsis have better nutrition status.

Specific comments:

  1. Abstract: The retrospective nature of the study should be clearly stated.

A: Thanks for your valuable comments. The sentence has been revised. (page1, line 19)

This retrospective study analysed the impact of body mass index (BMI) and modified Nutrition Risk in Critically ill (mNUTRIC) scores onsurvival of these patients.

Methods:

  1. Page 3. Line 110: A figure explaining the modified NUTRIC score would be of interest.

A: Thanks for your valuable comments. We added Table 1 to explain the modified NUTRIC score. (page3, Table 1)

Table 1. modified NUTRIC score

Variable

Range

Points

Age

<50

0

≥50 and <75

1

≥75

2

APACHE II

<15

0

≥15 and <20

1

≥20 and <28

2

≥28

3

SOFA

<6

0

≥6 and <10

1

≥10

2

Number of comorbidities

0-1

0

≥2

1

Days from hospital to ICU admission

≥0 and <1

0

≥1

1

  1. Page 3. Line 115-116: The authors have chosen 6 as their cut-off point in NUTRIC score based on the results of their sample. Is this congruent with previous literature?

A: Thanks for your valuable comments. Previous literature chose 5 as the cut-off point of modified NUTRIC score. According to reference 6, the case number of high mNUTRIC score (≥5) was 23, and low mNUTRIC score (<5) was 60. The number of patients with high mNUTRIC scores was only one-third of patients with low mNUTRIC scores. In our study, the case number of the high mNUTRIC score(≥ 6) was 513 and the low mNUTRIC score was 286. The patients with high mNUTRIC scores were almost twice as many as the patient with low mNUTRIC scores. It revealed our case had worse nutrition status compared to the past studies. To balance the number of subgroups, we chose 6 as the cut-off point. Even we chose 6 as the cut-off point, the survival of different mNUTRIC scores was consistent with past studies which chose 5 as cut-off point. 

  1. Page 3. Line 125: Biomarkers data were collected from day 1 after ICU admission. It would have been preferable to collect first sample on ICU admission to have baseline values. If this has not been possible, the authors should discuss on this shortcoming.

A: Thanks for your valuable comments. The first samples of biomarkers were collected on day 1 after ICU admission.

Results:

  1. Page 7. Line 255: The reference to figure 4b seems to be wrong. The authors refer in the text to the different 90-day survival rate between BMI groups when in line 248 the authors state the opposite. In addition, figure 4b refers to patients who had low mNUTRIC scores according to BMI.

A: Thanks for your valuable comments. We have revised the reference to figure 4b. (page 8, paragraph 1)

In the validation cohort, the 90-day survival rates were not significantly different among the different BMI groups with low mNUTRIC scores (Figure 4b).

Discussion:

  1. Page 14. Line 358: When referring to HbA1c levels, the authors should comment on which levels, low or high HbA1c levels predict the survival of patients with sepsis and low mNUTRIC scores?

A: Thanks for your valuable comments. We have revised the paragraph (page15, paragraph1). 

In this study, the 90-day survival curves showed that underweight patients with low mNUTRIC scores had significantly better survival rates than those with normal weights and low mNUTRIC scores (70.8% vs. 58.3%, respectively; p=0.048). In the re-gression analysis, being underweight and the HbA1c levels predicted the survival of patients with sepsis who had low mNUTRIC scores. 1 % increase of HbA1c had a trend to decrease 20% mortality in patients with sepsis and low mNUTRIC scores (ORR:0.792, p=0.078). Both these characteristics were protective factors for survival. Cancer was an independent risk factor for survival, and patients with cancer had a nearly four-times higher risk of mortality (βcancer =3.921; p<0.001) than those without cancer. Additionally, the mNUTRIC scores predicted 90-day survival rates in critically ill patients, but not in those with normal weights. This study revealed that when the mNUTRIC score of pa-tients with sepsis was ≤5, being underweight was a protective factor and an indicator of survivability. We also highlighted the limitations of the mNUTRIC scores of patients with sepsis who had normal weights.

  1. Page 16. Line 447: When discussing limitations, the authors should declare the retrospective nature of their design as one of the most important limitations of their study, and the need of further prospective trials to confirm their results. In addition, it is a single center study in which cut-off point regarding mNUTRIC score is obtained from your sample, that is external validity is likely low.

A: Thanks for your valuable comments. We have revised the paragraph (page17, paragraph1).

Nevertheless, this study had some limitations. First, the study used body weights measured during ICU admission. BMI might be misclassified because of a lack of fluid balance adjustment [23]. Second, BMI does not represent complete body components, especially body fat. Previous studies have shown that obesity may reduce the risk of mortality via the secretion of leptin and soluble TNF receptor 2 during sepsis in adipose tissues [265]. Third, finger sugar, HbA1c, and biological data were not available for the validation cohort. Further studies are needed to clarify the relationship between com-plete body components and sepsis. In addition, the survival of patients with sepsis may be affected by multiple factors, such as age, sex, comorbidity, infection sites, and anti-biotics. Although we used controls for age, sex, multiple comorbidities, and infection sites, we could not make a direct causal correlation between nutritional status and sur-vival, which would require further evidence. Since this was a single-center retrospective study, we need further prospective trials to confirm our results.

Conclusions:

Page 16. Lines 460-1: Authors should be careful in stating their conclusions. First, it is a retrospective study and second, as authors have correctly stated in their limitations, the survival of patients with sepsis may be influenced by multiple factors. Therefore, stating that their study reveals that being underweight was an indicator of survival in patients with sepsis who had low mNUTRIC scores <5 could be a leap of faith. Please make sure you synchronize your statements with your findings taking in account shortcomings.

A: Thanks for your valuable comments. We have revised our conclusion (page17, paragraph2).

The major contribution of our study was that it revealed that being underweight may not always be harmful if patients have optimal clinical nutritional status. Our findings have important implications for clinicians, as they can guide the application of thera-peutic strategies and provide a better understanding of the underlying mechanisms of sepsis-related mortality.

#Response of all reviewer: Please see the attachment.

Reviewer 3 Report

The paper by Tsai et al “Impact of body mass index on the survival of patients with sepsis with different modified NUTRIC scores” describes the results of the analysis of the impact of body mass index (BMI) and modified Nutrition Risk in Critically ill (mNUTRIC) score on the survival of patients with sepsis admitted to an intensive care unit. Sepsis is a serious health problem in intensive care patients worldwide and is associated with high mortality rates, so there is a strong need for useful information based on strong evidence to develop strategies to reduce this mortality.  The results of this study indicate that being underweight may be protective in patients with sepsis who had favourable mNUTRIC scores (≤5) and that diabetes may be associated with better outcomes. These results are in contrast with those of several other studies on the same topics, therefore a more detailed discussion and different conclusions are strongly needed.

In particular, as regards the underweight, in the discussion starting from line 368, the authors cite five studies showing that being underweight may be a risk factor for patients with sepsis and one study in accord with their results, even if they stated: “Previous studies showed that being underweight may be a risk factor for patients with sepsis [18–22], while others did not find a significant association between being underweight and outcomes in these patients [40].”; thus “others” must be changed with “one”. Moreover, the studies that demonstrated a relationship between being underweight and an increased risk of sepsis-related mortality are by far more than five, as an example above all a recent study on this topic with a very large cohort, such as 0.5 million Chinese adults, is not cited (doi: 10.1186/s13054-020-03229-2). For these reasons it is risky to consider a strength of the present study the fact that it revealed that being underweight was an indicator of survivability in patients whose mNUTRIC scores were ≤5, starting from a sub-group of 64 underweight subjects with better other risk parameters than the other BMI subgroups studied.

All the above concerns about overweight are also applicable for the potential protective effect of diabetes suggested by the present study.

Author Response

Many Thanks for your valuable comments and suggestions which have led to significant improvement on the presentation and quality of this paper. In what follows, we shall detail the changes we have made on the paper.

Reviewer

The paper by Tsai et al “Impact of body mass index on the survival of patients with sepsis with different modified NUTRIC scores” describes the results of the analysis of the impact of body mass index (BMI) and modified Nutrition Risk in Critically ill (mNUTRIC) score on the survival of patients with sepsis admitted to an intensive care unit. Sepsis is a serious health problem in intensive care patients worldwide and is associated with high mortality rates, so there is a strong need for useful information based on strong evidence to develop strategies to reduce this mortality.  The results of this study indicate that being underweight may be protective in patients with sepsis who had favourable mNUTRIC scores (≤5) and that diabetes may be associated with better outcomes. These results are in contrast with those of several other studies on the same topics, therefore a more detailed discussion and different conclusions are strongly needed.

In particular, as regards the underweight, in the discussion starting from line 368, the authors cite five studies showing that being underweight may be a risk factor for patients with sepsis and one study in accord with their results, even if they stated: “Previous studies showed that being underweight may be a risk factor for patients with sepsis [18–22], while others did not find a significant association between being underweight and outcomes in these patients [40].”; thus “others” must be changed with “one”. Moreover, the studies that demonstrated a relationship between being underweight and an increased risk of sepsis-related mortality are by far more than five, as an example above all a recent study on this topic with a very large cohort, such as 0.5 million Chinese adults, is not cited (doi: 10.1186/s13054-020-03229-2). For these reasons it is risky to consider a strength of the present study the fact that it revealed that being underweight was an indicator of survivability in patients whose mNUTRIC scores were ≤5, starting from a sub-group of 64 underweight subjects with better other risk parameters than the other BMI subgroups studied.

All the above concerns about overweight are also applicable for the potential protective effect of diabetes suggested by the present study.

A: Thanks for your valuable comments. We have added the recent study: “Body-mass index and long-term risk of sepsis-related mortality: a population-based cohort study of 0.5 million Chinese adults” in reference 24 and revised our discussion (page 15, paragraph 1. 2) and conclusion (page17, paragraph2).

Discussion (page 15, paragraph 1. 2)

In this study, the 90-day survival curves showed that underweight patients with low mNUTRIC scores had significantly better survival rates than those with normal weights and low mNUTRIC scores (70.8% vs. 58.3%, respectively; p=0.048). In the re-gression analysis, being underweight and the HbA1c levels predicted the survival of patients with sepsis who had low mNUTRIC scores. 1 % increase of HbA1c had a trend to decrease 20% mortality in patients with sepsis and low mNUTRIC scores (ORR:0.792, p=0.078). Both these characteristics were protective factors for survival. Cancer was an independent risk factor for survival, and patients with cancer had a nearly four-times higher risk of mortality (βcancer =3.921; p<0.001) than those without cancer. Additionally, the mNUTRIC scores predicted 90-day survival rates in critically ill patients, but not in those with normal weights. This study revealed that when the mNUTRIC score of pa-tients with sepsis was ≤5, being underweight was a protective factor and an indicator of survivability. We also highlighted the limitations of the mNUTRIC scores of patients with sepsis who had normal weights.

The present study revealed that being underweight was associated with better survival in patients with sepsis who had low mNUTRIC scores. Previous studies showed that being underweight may be a risk factor for patients with sepsis [18–22, 24], while one study did not find a significant association between being underweight and outcomes in these patients [41]. Nevertheless, these studies did not consider the heterogeneity of patients with sepsis in the same BMI group. Most of them lacked baseline characteristics of clinical scores (NUTRIC score, SOFA score, and APACHE II) and laboratory data (HbA1C and blood glucose). In our analysis, the overall survival of patients with sepsis was not significantly different from that of those with normal weights. However, there were differences in the results of the subgroup analysis. Our results showed that better nutritional status of patients with sepsis was correlated with improved survivability in underweight patients. We also found that patients with low mNUTRIC scores had the lowest SOFA scores on days 1, 3, and 7 among the three BMI groups.The SOFA scores were used to evaluate the severity of sepsis and predict the outcomes of patients with sepsis [1]. In model 1 of Table 4, the underweight group had a trend to have better survival in patients with sepsis (0.557, p=0.082). However, in adjusted model 2, the trend disappeared after we adjusted it by SOFA scores. Thus, the survival of the underweight patients might have been indicated by the low SOFA scores. Previous studies showed underweight patients had lower SOFA scores compared to normal weight and obesity patients [42,43]. However, the underweight group with lower SOFA scores didn’t have better survival in past studies. It may be related to heterogenecity of the underweight group. Unlike previous studies, we categorized patients with sepsis by mNUTRIC score which was used to differentiate nutrition status and severity. Our results revealed that underweight might not be a harmful factor when patients with sepsis have better nutrition status.

Conclusion (page17, paragraph2).

The major contribution of our study was that it revealed that being underweight may not always be harmful if patients have optimal clinical nutritional status. Our findings have important implications for clinicians, as they can guide the application of thera-peutic strategies and provide a better understanding of the underlying mechanisms of sepsis-related mortality.

# response of all reviewers: Please see the attachment.

Round 2

Reviewer 3 Report

Reviewing the manuscript the authors met my requests